# Targeted protein degradation by KLHDC2 ligands identified by high-throughput screening

Han Zhou[1†], Tonglian Zhou[1†], Wenli Yu[2], Liping Liu[1], Yeonjin Ko[3], Kristen A Johnson[4], Ian A Wilson[2], Peter G Schultz[1], Michael J Bollong[1]*

[1]Department of Chemistry, Scripps Research, La Jolla, United States; [2]Department of Integrative Structural and Computational Biology, Scripps Research, La Jolla, United States; [3]Chemical & Biological Integrative Research Center, Korea Institute of Science and Technology, Seoul, Republic of Korea; [4]Department of Biology, Calibr-Skaggs at Scripps Research, La Jolla, United States

*For correspondence:
mbollong@scripps.edu

[†]These authors contributed equally to this work

Competing interest: The authors declare that no competing interests exist.

## eLife Assessment

This **valuable** study aims to advance the toolkit of small molecules used for approaches to targeted protein degradation for research and therapeutic applications. The authors provide **solid** data demonstrating the use of a high-throughput screen of small molecules to target a specific E3 ligase, KLHDC2 (Kelch-like homology domain containing protein 2); the resulting compounds then form the basis for new PROTAC (proteolysis targeting chimera) reagents. The strength of the work lies in expanding the PROTAC reagent inventory. The current work would be strengthened further by confirming that the PROTAC's activity is dependent on KLHDC2 and by a more thorough examination of off-target effects in cellular applications.

**Abstract** Proteolysis-targeting chimeras (PROTACs) enable the selective and sub-stoichiometric elimination of pathological proteins, yet only two E3 ligases are routinely used for this purpose. Here, we expand the repertoire of PROTAC-compatible E3 ligases by identifying a novel small molecule scaffold targeting the ubiquitin E3 ligase KLHDC2 using a fluorescence polarization-based high-throughput screen. We highlight the utility of this ligand with the synthesis of PROTACs capable of potently degrading BRD4 in cells. This work affords additional chemical matter for targeting KLHDC2 and suggests a practical approach for identifying novel E3 binders by high-throughput screening.

## Introduction

Targeted protein degradation has emerged as an alternative strategy to modulate cellular signaling. Among established pharmacological approaches, proteolysis-targeting chimeras (PROTACs) have gained prominence in the field, enabling the potent degradation of several challenging cancer targets like estrogen receptor, androgen receptor, among others, with several progressing to ongoing clinical trials (*Békés et al., 2022*; *Teng and Gray, 2023*; *Berkley et al., 2025*). PROTACs are heterobifunctional molecules composed of a ligand targeting the protein of interest, a ligand for a ubiquitin E3 ligase, and a linker with precise geometry to bring the two proteins within sufficient proximity to induce the ubiquitination and degradation of the so-called 'neosubstrate' proteins (*Sakamoto et al., 2001*). Despite the widespread clinical and academic use of PROTACs, most degrader molecules developed to date have exploited only two E3 ligases, VHL (von Hippel-Lindau) and CRBN (Cereblon)

**Figure 1.** A fluorescence polarization-based screen identifies ligands of the Kelch domain of KLHDC2. (**A**) Structure of the Kelch domain of KLHDC2 from PDB 6DO3 bound to PPPMAGG, C-terminal peptide from SelK (left). Interface of SelK peptide (teal) bound to KLHDC2 (right) with TAMRA-labeled peptide used in this work above. (**B**) Fluorescence polarization signal of TAMRA-SelK peptide in response to increasing concentrations of GST-KLHDC2. (**C**) Fluorescence polarization signal from KLHDC2 and KEAP1 assays in response to increasing concentrations of unlabeled SelK peptide (n=3; mean and s.e.m.). Fluorescence polarization signals in Z'-based determination assays with and without unlabeled SelK peptide (1 μM) in 384-well (**D**) and 1536-well (**E**) assays (n=20). (**F**) Corrected fluorescence polarization signal from the primary screening campaign with hits noted in teal. (**G**) Screening funnel depicting the high-throughput screening campaign.

The online version of this article includes the following source data and figure supplement(s) for figure 1:

**Source data 1.** Raw data corresponding to *Figure 1B–F*.

**Figure supplement 1.** A fluorescence polarization-based control assay with Kelch domain-containing protein KEAP1.

---

(*Buckley et al., 2012*; *Winter et al., 2015*). While several additional E3 ligases have been reported in the literature as being PROTAC-compatible, these ligands do not have well-understood mechanisms of E3 engagement and often do not fully degrade their target substrates (*Ishida and Ciulli, 2021*). As such, an expanded repertoire of E3 ligands would likely enable the degradation of a broader array of protein targets to endow subcellular and/or tissue specificity of degradation or to overcome potential resistance mutations in currently targeted E3s.

Among potential E3 ligases for PROTAC development, KLHDC2 (Kelch-like homology domain-containing protein 2) has emerged as a promising candidate (*Békés et al., 2022*). A CUL2 adaptor protein, KLHDC2, serves as a key component of the C-end degron pathway, recognizing C-terminal diglycine residues of client proteins, promoting their ubiquitination and degradation (*Pilcher et al., 2025*). Client proteins of KLHDC2 include selenocysteine-containing proteins, like SelK (Selenoprotein K), which, in the context of selenocysteine depletion, undergo premature translational termination, revealing a C-terminal diglycine degron (*Rusnac et al., 2018*). Previous work has shown that recognition of the diglycine degron of SelK by KLHDC2 in its Kelch beta propellor domain is characterized by low nanomolar dissociation constants (<10 nM) and with favorable binding kinetics for PROTAC development (*Figure 1A*; *Rusnac et al., 2018*). Unbiased protein proximity-based screens for generalizable E3 degraders have also indicated that KLHDC2 is among the best E3 ligases for fully degrading generic protein substrates (*Poirson et al., 2024*; *Röth et al., 2023*). Similarly, others have shown that peptide-based PROTACs derived from a 7-mer peptide of the C-terminus of SelK can be used to degrade a variety of kinase targets in human cells (*Kim et al., 2023*).

We reasoned that the well-characterized interaction between KLHDC2 and SelK could be exploited to identify small molecule binders of KLHDC2 from a competitive in vitro screen using recombinant

protein. Accordingly, we report here the development and execution of an FP-based high-throughput screen that identified a tetrahydroquinoline-based scaffold that competes for SelK peptide binding. We demonstrate this scaffold can be optimized for potency, achieving submicromolar affinity for KLHDC2 binding and can be further derivatized with JQ1 to degrade BRD4 in cell-based assays.

## Results

To identify small molecules capable of binding the Kelch domain of KLHDC2, we first established a miniaturized fluorescence polarization (FP)-based assay reporting on the KLHDC2-SelK interaction. This assay derives from literature precedent in which a C-terminal peptidic fragment of SelK (HLRG-SPPPMAGG) has been shown to potently associate with purified GST-tagged KLHDC2 in a recombinant AlphaScreen assay (*Rusnac et al., 2018*). Accordingly, we evaluated fluorescence polarization signal in response to increasing concentrations of recombinantly produced GST-tagged KLHDC2 from Sf9 insect cells in the presence of a 12-mer SelK peptide N-terminally conjugated with a TAMRA fluorophore (3.1 nM, TAMRA-HLRGSPPPMAGG). Under these conditions, a slightly shifted dissociation constant was achieved with the fluorophore-tagged SelK peptide (25 nM, *Figure 1B*) relative to the reported literature value of 3.4 nM for the unlabeled peptide (*Rusnac et al., 2018*). We then demonstrated that unlabeled SelK peptide can compete for KLHDC2 binding ($IC_{50}$=55 nM) under these conditions, giving us good confidence in recapitulating this known interaction and providing a positive assay control for high-throughput screening (*Figure 1C*). A suitable 384-well FP assay with GST KLHDC2 (25 nM) and the TAMRA-tagged SelK peptide (3.1 nM) was achieved (Z'=0.81) when exposed to a molecular excess of free SelK peptide (10 µM; *Figure 1D*).

The FP assay was further miniaturized to 1536-well format (Z'=0.61, *Figure 1E*) and then screened against a subset of the Calibr small molecule library (354,274 compounds) using automated high-throughput screening for compounds which decreased the polarization signal (*Figure 1F*). From ~14,000 primary screening hits, 51 displayed dose-responsive inhibitory signals in follow-up

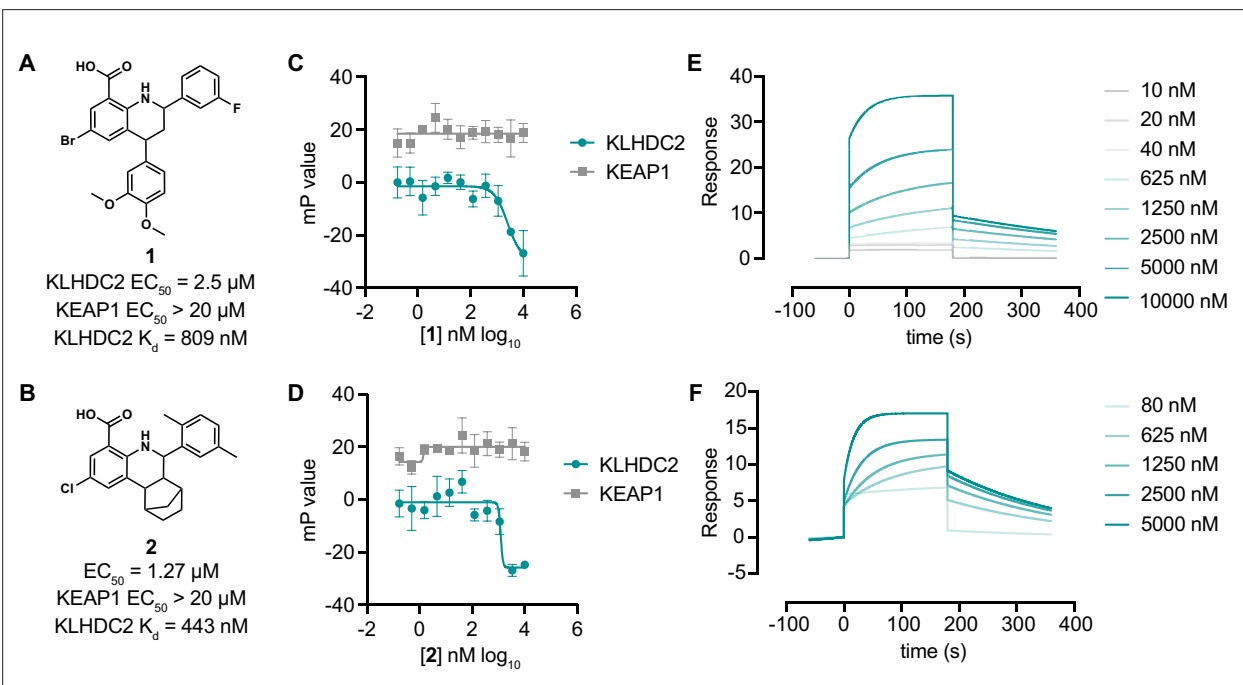

**Figure 2.** Tetrahydroquinoline-based KLHDC2 ligands. Structures and summary of activities of compounds **1** (**A**) and **2** (**B**). (**C**) Fluorescence polarization signal from KLHDC2 and KEAP1 assays in response to increasing concentrations of **1** (**C**) and **2** (**D**) (n=3; mean and s.e.m.). Representative surface plasmon resonance (SPR) sensorgrams of KLHDC2 binding from the indicated concentrations of **1** (**E**) and **2** (**F**).

The online version of this article includes the following source data and figure supplement(s) for figure 2:

**Source data 1.** Raw data corresponding to *Figure 2C–F*.

**Figure supplement 1.** A 2-amino thiazole-based KLHDC2 ligand scaffold.

assays. After attriting these molecules to remove potential fluorophores and quenchers, we ultimately identified four validated hit compounds that dose-dependently decreased FP signal with half-maximal inhibitory concentrations in the single-digit micromolar range (*Figure 1G*).

In parallel, we devised a counter screen to control for nonspecific binders using a different Kelch domain-containing ubiquitin E3 ligase, KEAP1 (Kelch-like ECH-associated protein 1), which has also been co-opted for targeted protein degradation using PROTACs (*Du et al., 2022*). Using recombinant KEAP1 Kelch domain expressed from *Escherichia coli*, we were able to establish a dissociation constant ($K_d$ = 22 nM) via FP with a peptide corresponding to the KEAP1 substrate NRF2 (NFE2L2), which was N-terminally tagged with a TAMRA fluorophore (TAMRA-AFFAQLQLDEETGEFL, *Figure 1—figure supplement 1A*). We confirmed this FP signal could be competed with good potency ($IC_{50}$=6 nM) by a previously reported small molecule ligand of the KEAP1 Kelch domain, KI-696 (*Figure 1—figure supplement 1B and C*). Importantly, the free SelK peptide displayed no inhibitory signal in FP assays measuring the association between KEAP1 and NRF2 (*Figure 1C*), further validating the specificity of the SelK-KLHDC2 FP-based interaction assay used for screening.

We then characterized the four hits for their capacity to bind to KLHDC2 in vitro. Two of these hits, **1** and **2**, share a conserved 8-carboxy tetrahydroquinoline scaffold (*Figure 2A and B*), whereas the other two compounds, **3** and **4**, were closely related imidazopyridines that varied only by an additional methyl substitution present in **4** (*Figure 2—figure supplement 1*). The four compounds displayed dose-dependent inhibitory effects in the SelK-KLHDC2 competitive FP assay in 384-well format ($IC_{50}$s=2.5 μM for **1**, 1.3 μM for **2**, 3.6 μM for **3**, and 2.8 μM for **4**) and did not have an inhibitory effect in the KEAP1 counter-screening assay at concentrations less than 20 μM (*Figure 2C and D*; *Figure 2—figure supplement 1C, D*). We next evaluated these four molecules for direct binding to immobilized KLHDC2 using surface plasmon resonance (SPR). **1** and **2** gave robust interaction responses, yielding $K_d$ values of 810 nM and 440 nM, respectively (*Figure 2E and F*). Compounds **3** and **4** were not found to provide robust sensorgram signals. Whether this observation derives from incompatibility under the assay conditions or requires the presence of the SelK peptide for interaction with KLHDC2 remains unclear. As such, we chose to evaluate the tetrahydroquinoline series for further study.

We next sought to understand if the tetrahydroquinoline scaffold might be further optimized for increased affinity with KLHDC2. We evaluated all commercially available analogs that shared the 8-carboxy tetrahydroquinoline core (53 analogs) in 384-well FP assays measuring association between

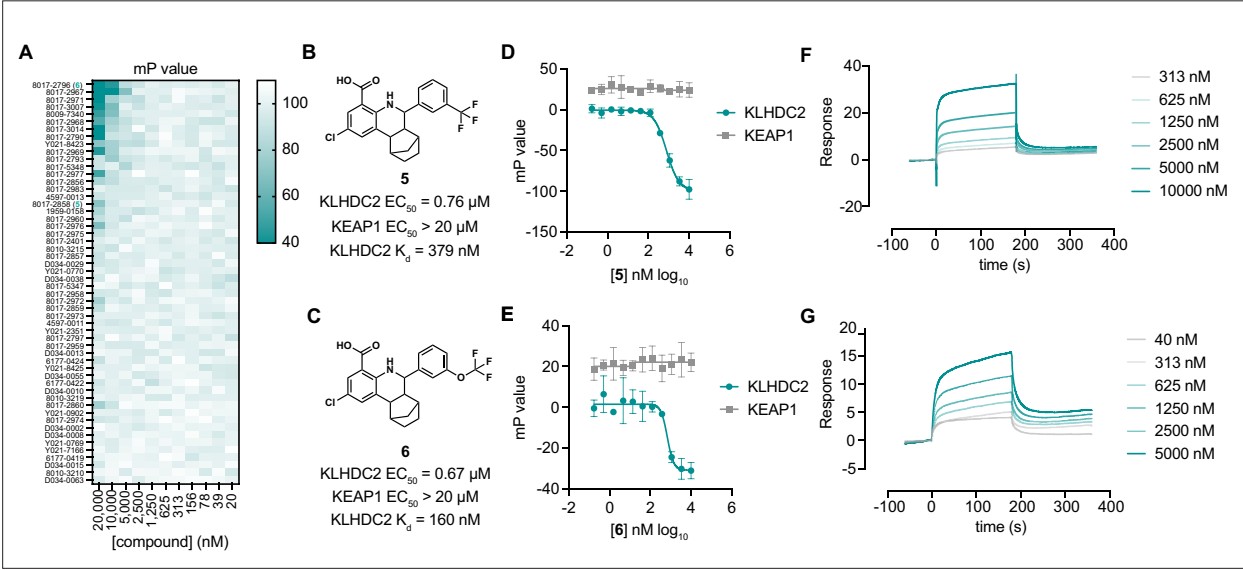

**Figure 3.** Optimization of tetrahydroquinolines as KLHDC2 ligands. (**A**) Heatmap of fluorescence polarization signal in response to dose responses of the 54 related compounds. Structures and summary of activities of compounds 5 (**B**) and 6 (**C**). Fluorescence polarization signal from KLHDC2 and KEAP1 assays in response to increasing concentrations of 5 (**D**) and 6 (**E**) (n=3; mean and s.e.m.). Representative surface plasmon resonance (SPR) sensorgrams of KLHDC2 binding from the indicated concentrations of 5 (**F**) and 6 (**G**).

The online version of this article includes the following source data for figure 3:

**Source data 1.** Raw data corresponding to *Figure 3A, D–G*.

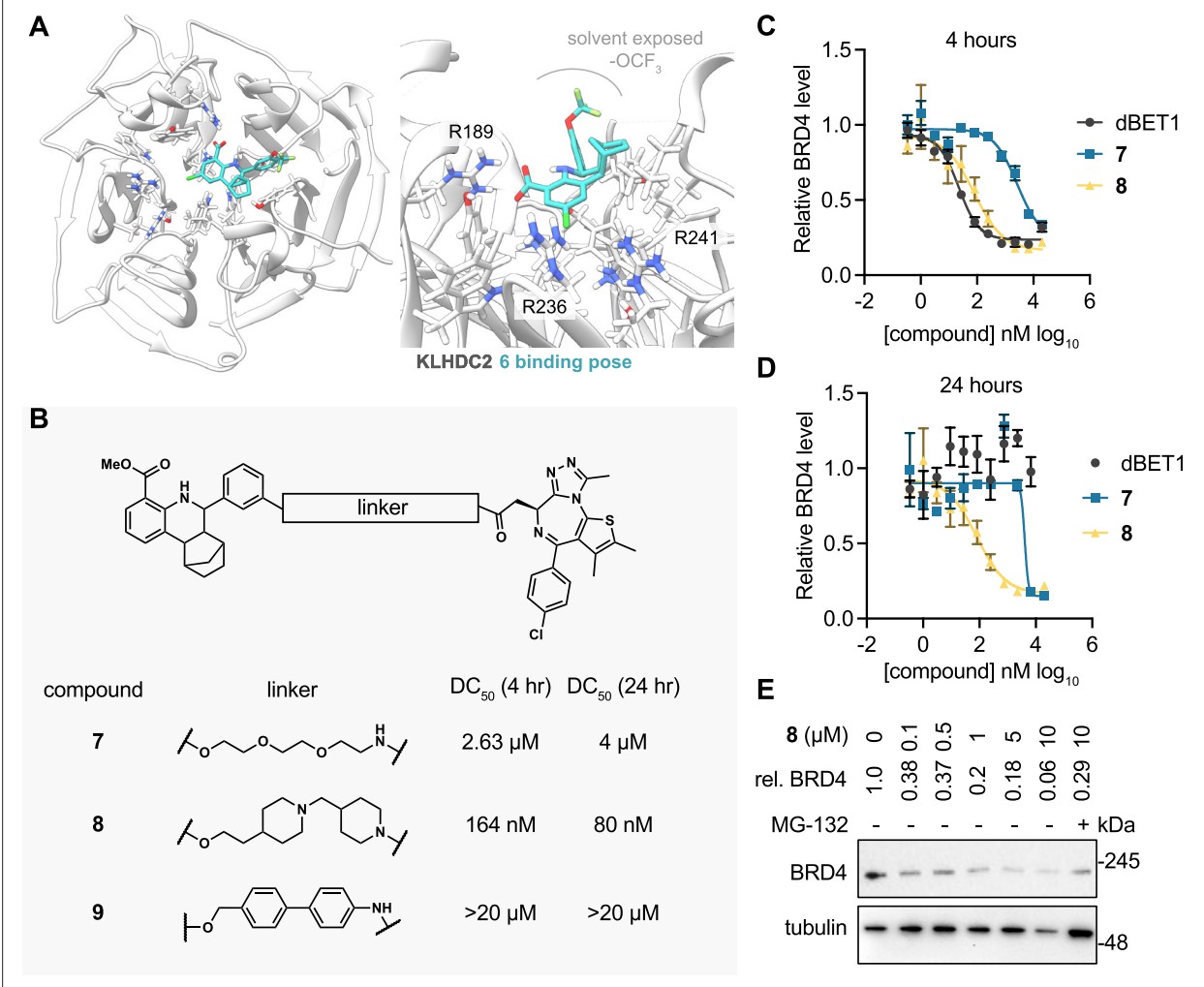

**Figure 4.** KLHDC2 ligands can be co-opted for targeted degradation of BRD4. (**A**) Binding pose of compound **6** to KLHDC2 (PDB 6DO3). (**B**) Structure and summary of activities of the indicated proteolysis-targeting chimeras (PROTAC) molecules. Relative BRD4 levels at 4 (**C**) or 24 hr (**D**) as measured by luminescence from HEK293T cells expressing a HiBiT-tagged BRD4 transgene (n=3; mean and s.e.m.). (**E**) Representative western blot and densitometry-based quantification of BRD4 levels from SK-BR-3 in response to 24 hr treatment with the indicated concentrations of **8**.

The online version of this article includes the following source data for figure 4:

**Source data 1.** Raw data corresponding to *Figure 4C and D*.

**Source data 2.** Uncropped blots of *Figure 4E*.

**Source data 3.** Original image scans of *Figure 4E*.

KLHDC2 and the labeled SelK peptide. Notably, substitution of 2,5-dimethyl substituted phenyl group present in **2** to 3-substituted trifluoromethyl or trifluoromethoxy groups was found to increase potency in this initial screen (*Figure 3A*). We evaluated the trifluoromethyl substituted analog **5** and the trifluoromethoxy substituted **6** (*Figure 3B and C*) in follow-up FP assays with KLHDC2, observing increased potency for these analogs ($IC_{50}$s=760 nM and 670 nM respectively, *Figure 3D and E*) with no inhibitory effect in the KEAP1 counter-screening assay. Compounds **5** and **6** also displayed increased potency in SPR assays measuring affinity for KLHDC2 binding with $K_d$s of 380 nM and 160 nM, respectively (*Figure 3F and G*).

We next sought to determine if the tetrahydroquinoline series could be used for targeted protein degradation. We performed docking studies using AutoDock Vina to evaluate the potential binding modes of **6** to the previously solved crystal structure of the Kelch domain of KLHDC2 (PDB 6DO3) (*Rusnac et al., 2018*). Surprisingly, **6** was not predicted to be capable of making interactions with Arg236 or Arg241, which recognize the anionic carboxy terminus of the C-terminal diglycine motif

of SelK, as anticipated. Instead, **6** occupies a more distal, lipophilic spot in the SelK binding pocket, making a predicted H-bond interaction with Arg189 of KLHDC2 (*Figure 4A*). In this orientation, the trifluoromethoxy group points to solvent, providing a potential vector for derivatizing PROTAC linkers. We evaluated this hypothesis by synthesizing three PROTAC molecules bearing JQ1, a well-characterized ligand to BRD4 (bromodomain-containing protein 4), which has been used often in the literature as a means of testing PROTAC activity (*Winter et al., 2015*; *Lu et al., 2015*). These molecules were based on the methyl ester substituted scaffold represented by **2** (for increased cellular permeability and subsequent hydrolysis of the carboxylate by cellular esterases) and bore 3 position substituted PROTAC linkages to JQ1. These PROTACs linkers consisted of the flexible three ethylene glycol substituted **7** as well as the more structured linkers represented by **8** and **9** (*Figure 4B*). We evaluated the ability of these molecules to degrade BRD4 using a HEK293T-based assay in which the luminescence produced from a transiently transfected BRD4-HiBiT transgene correlates to the amount of cellular BRD4 present. We found **7** degraded BRD4 in the BRD4-HiBiT with modest activity after 4 hr of treatment (half-maximal degradation concentration, $DC_{50}=2.6$ µM) and that degradation by this molecule persisted for 24 hr ($DC_{50}=4$ µM; *Figure 4C and D*). Notably, the more rigid PROTAC **8** displayed considerably enhanced activity in degrading BRD4 within 4 hr of treatment ($DC_{50}=164$ nM) that further increased ($DC_{50}=80$ nM) at the 24 hr timepoint (*Figure 4C and D*). This result contrasts with dBET1, a reported CRBN-targeting PROTAC degrader of BRD4 (*Winter et al., 2015*), which potently promoted degradation at 4 hr in this assay ($DC_{50}=22$ nM) but became inactive at the 24 hr timepoint. Notably, the most rigid biphenyl linker containing **9** did not promote BRD4 degradation at any concentration (<20 µM) or timepoint (4 or 24 hr) tested. We lastly confirmed the ability of **8** to degrade endogenous BRD4 in SK-BR-3 cells. **8** (100 nM) degraded more than 60% of endogenous BRD4 protein in 24 hr of treatment; maximal degradation was observed with the highest concentration tested (10 µM; 94% degradation; *Figure 4E*). Importantly, the 10 µM condition could be partially rescued when co-treated with MG-132 (29% remaining vs. 4% in the DMSO condition), indicating degradation occurs through a proteasome-mediated mechanism.

## Discussion

Here, we have reported the development and execution of a high-throughput competitive FP-based screen that identified a novel ligand to KLHDC2, an E3 ligase effector of the C-end degradation rule. One tetrahydroquinoline scaffold was found to dose-dependently bind KLHDC2 in two in vitro assays, consistent with a binding mode that competes for interaction with KLHDC2 client proteins in the SelK recognition pocket. Importantly, we were able to further optimize the affinity of this series by several fold, yielding **6**, a ligand which bound recombinant KLHDC2 in SPR assays with a $K_d$ of 160 nM. As such, we hypothesize that **6** and its analogs will be useful tools to further probe the biological roles of KLHDC2 in cells. Recently, several other groups have used computational design to identify ligands to KLHDC2 (*Scott et al., 2024*; *Zhou et al., 2024*; *Hickey et al., 2024*). These ligands engage the diglycine recognition motif deep within the SelK binding pocket of KLHDC2, which contrasts with the predicted binding mode of **6** presented here. We demonstrated that the tetrahydroquinoline scaffold can be further derivatized, yielding PROTAC molecules like **8**, which can degrade BRD4 in cells with long-acting degradation kinetics. Like other PROTAC designs, linker geometry is important, with the semirigid linker of **8** considerably outperforming other designs. To the extent that **8** may be further modified to increase its potency and drug likeness will be the subject of future work. To our knowledge, this is the first demonstration that a noncovalent, PROTAC-compatible E3 ligase ligand can be discovered by affinity-based, high-throughput screening. Overall, this work adds a novel scaffold for targeting KLHDC2 and further supports the notion that drug-like molecules can be developed to this E3 for therapeutic applications.

## Materials and methods
### Cell culture

HEK293T and SK-BR-3 cells were purchased from American Type Culture Collection (ATCC) and were profiled for identity using Short Tandem Repeat (STR) profiling. HEK293T cells were maintained in DMEM (Gibco) containing 10% fetal bovine serum (Gibco) and 1% penicillin streptomycin (Gibco).

SK-BR-3 cells were maintained in McCoy's 5A medium (Gibco) supplemented with 10% fetal bovine serum (Gibco) and 1% penicillin streptomycin (Gibco).

## Protein expression and purification

The Kelch domain of human KLHDC2 (amino acids 1–362) was cloned in frame with an N-terminal glutathione-*S*-transferase (GST) tag and TEV-cleavage sequence into the pFastBac vector and was expressed in two rounds of baculoviral production in insect Sf9 monolayer cells. The produced 15 mL of P3 virus was used to infect 1.5 L HiFive suspension cells at a cellular density of $10^6$/mL in a 27°C shaker at 110 RPM. After 3 days of infection, the cells were harvested at 8000×$g$ for 30 min and then lysed in 20 mM Tris, pH 8.0, 200 mM NaCl, 5 mM DTT in the presence of protease inhibitors (1 mg/mL leupeptin, 1 mg/mL pepstatin, and 100 mM PMSF) with sonication. The lysate was incubated with Pierce Glutathione Agarose (Thermo Scientific) at 4°C overnight and eluted with 250 mM glutathione. The crude protein was first dialyzed into dialysis buffer (20 mM Tris, pH 8.0, 200 mM NaCl, 5 mM DTT) and further purified with size exclusion using a Superdex-75 column (GE Healthcare). The human KEAP1 Kelch domain was expressed using a pET21a-KEAP1 plasmid outfitted cloned in frame with an N-terminal $HIS_6$ tag. Plasmid-containing BL21(DE3) cells were amplified in 1 L 2YT culture at 37°C until the $OD_{600}$ reached 0.8, followed by 1 mM IPTG induction at 4°C overnight. The harvested cells were lysed in the aforementioned lysis buffer and purified via a standard Ni-NTA purification protocol. Both proteins were concentrated with an Amicon concentrator (Sigma), and the protein concentrations were determined by BCA assay (Thermo Fisher 23327). All protein samples were flash-frozen in liquid nitrogen for future use.

## Small molecule libraries

The small molecule library of 354,274 compounds consisted of a ChemDiv library (150,114 compounds), an Enamine library (142,208 compounds), a Life Chemicals library (33,792 compounds), and the ReFrame library (28,160 compounds).

## High-throughput screening

Compounds from selected small molecule libraries were pre-spotted at 10 µM (final concentration) with Echo Acoustic liquid handler system (Beckman) in the format of 1536-well or 384-well in Greiner solid black microplates (Cat.No. 782076 and 781209). The C-terminal peptide of SelK, native substrate KLHDC2 protein, HLRGSPPPMAGG (InnoPep. Inc), was spotted into each individual plate, which serves as positive control at a final concentration of 1 µM. The FP assay was carried out by first dispensing 5 µL of KLHDC2 protein at 25 nM for 1536-well microplates or 25 µL at 25 nM for 384-well microplates with a Multidrop Dispenser (Thermo Fisher) followed by incubation at room temperature for 1 hr. An equal volume of 3.12 nM TAMRA-HLRGSPPPMAGG (InnoPep. Inc) was dispensed into the plates and incubated at room temperature for another 1 hr in the dark. The final concentration of the protein and the corresponding TAMRA-peptide was 12.5 nM and 1.56 nM, respectively. FP signals were recorded using an Envision plate reader (Perkin Elmer), and the robust Z-scores were calculated using Genedata Screener where ~3% of the hits were selected each batch based on robust Z-scores for the primary screening. The second round of triplicate validation and the third round of dose-response characterization were carried out in a similar format.

## Fluorescence polarization assays

The in-house validation of KLHDC binders including re-purchased SAR by inventory compounds was carried out in black solid-bottom 384-well microplates. 25 µL of KLHDC2 or KEAP1 protein was first dispensed into the plates, followed by transferring the corresponding concentration of compound through the Bravo Automated liquid handling system (Agilent). The mixture was incubated at room temperature for 1 hr before the TAMRA-peptide was added. The final concentration of the protein and the corresponding TAMRA-peptide was 25 nM and 3.12 nM, respectively. The plates were incubated at room temperature in a shaded environment for 1 more hour, and the FP signals were measured by Envision plate reader (Perkin Elmer). The data were analyzed and visualized by GraphPad Prism software.

## SPR binding experiments with KLHDC2$^{Klech}$

10 µM His-GST-KLHDC2 in 100 µL of 1X HBS (HEPES-buffered saline, 10 mM HEPES, 150 mM NaCl, pH 7.4) containing 25 µM EZ-Link NHS-PEG4-biotin was incubated at 4°C for 2 hr, after which the

reaction was quenched with 2 μL of 1 M Tris pH 7.5. The solution was dialyzed over an 18 hr period against three changes of 500 mL of 1X HBSS buffer. Biotinylated-His-GST-KLHDC2 was immobilized onto Xantec High Density SA chip by diluting to 0.7 μM injected over the surface for 60 s at 10 μL/min and resulting in approximate immobilized signal gain of 4000–4500 response units (RU). The running buffer for all immobilization assays was 50 mM Tris pH 8, 250 mM NaCl, 0.05% Tween 20, 0.1 mM DTT, and 2% DMSO. All measurements of direct binding in SPR experiments were collected using the Biacore 8K+ instrumentation.

### Cell-based HiBIT BRD4 degradation assay

HEK293T cells ($0.2 \times 10^6$ cells/well, 40 μL) were seeded in white opaque bottom Greiner 384-well microplates for 24 hr before transfection with 1 ng pCMV6-HiBIT-BRD4 plasmid, 99 ng pUC19 plasmid, and 4 μL/μg FuGene (Promega) in OptiMEM (Thermo Fisher). The next day, compounds were spotted as DMSO solutions (100 nL) using a Bravo Automated Liquid Handler (Agilent) affixed with a pintool head (V&P Scientific) in serial dilution. After 4 hr of incubation, 30 μL of Nano-Glo HiBIT Lytic Reagent was added, and the plates were shaken for 5 min. The HiBIT-BRD4 levels of each well were measured by Envision plate reader (Perkin Elmer). The raw data was analyzed and visualized by GraphPad Prism software.

### Cell-based endogenous BRD4 degradation assay

SK-BR-3 cells ($0.25 \times 10^6$ cells/well, 2 mL) were seeded in six-well polystyrene plates for 24 hr before the compounds were added at the indicated doses as DMSO solutions. The protease inhibitor MG132 (Selleck Chemicals) was added to the sixth well along with 10 μM 3d at a concentration of 10 μM. The plate was incubated at 37°C for 24 hr, and the cells from each well were collected and lysed in RIPA buffer (Thermo Fisher 89900) with Halt protease and phosphatase inhibitor cocktail (Thermo Fisher 78440). The cellular content was extracted on ice for 20 min, and the protein concentration was determined by BCA assay (Thermo Fisher 23327). 2 μg of each sample was resolved via SDS-PAGE gels, and western blotting was used to visualize the endogenous BRD4 level (BRD4 primary antibody Abcam plc, Ab128874, RRID:AB_11145462, 1:10,000 dilution) with Bio-Rad Gel Doc XR+ imaging system.

### Statistics

All experiments were performed at least twice. All data are technical replicates.

### Acknowledgements

This work was supported by the NIH (GM146865 to MJB and GM145323 to PGS) and Calibr. We would like to thank Savni Prabhu and Phillip Ordoukhanian for generating SPR data.

## Additional information

### Funding

| Funder | Grant reference number | Author |
|---|---|---|
| National Institute of General Medical Sciences | GM146865 | Michael J Bollong |
| National Institute of General Medical Sciences | GM145323 | Peter G Schultz |

The funders had no role in study design, data collection and interpretation, or the decision to submit the work for publication.

### Author contributions

Han Zhou, Data curation, Formal analysis, Investigation, Methodology, Writing - original draft; Tonglian Zhou, Liping Liu, Yeonjin Ko, Data curation, Methodology; Wenli Yu, Methodology; Kristen A Johnson, Resources, Methodology, Project administration; Ian A Wilson, Peter G Schultz, Resources, Project

administration, Writing - review and editing; Michael J Bollong, Conceptualization, Formal analysis, Writing - original draft, Project administration, Writing - review and editing

**Author ORCIDs**
Michael J Bollong ⬤ https://orcid.org/0000-0001-9439-1476

Reviewer #1 (Public review): https://doi.org/10.7554/eLife.106844.2.sa1
Reviewer #2 (Public review): https://doi.org/10.7554/eLife.106844.2.sa2

# Additional files

## Supplementary files
MDAR checklist

## Data availability
All data generated in this study are included in the manuscript and source data files.

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

## Appendix 1

### Synthetic methods and characterization

**Appendix 1—scheme 1.** Synethic scheme of the synthesis of Compound 7.

## Methyl 2-chloro-6-(3-hydroxyphenyl)-5,6,6a,7,8,9,10,10a-octahydro-7,10-methanophenanthridine-4-carboxylate (S4)

Methyl 2-amino-5-chlorobenzoate S1 (1.85 g, 10 mmol), norbornene S2 (1.41 g, 15 mmol), and 3-hydroxybenzaldehyde S3 (1.34 g, 11 mmol) were dissolved in DCM (50 mL) in a 200 mL Ace pressure tube. After purging with argon, boron trifluoride etherate (284 mg, 2 mmol) was added. The tube was sealed instantly, and the reaction mixture was heated at 40°C for 16 hr. 1 mL of methanol was added to the mixture to quench the reaction. After concentrating in vacuo, the crude residue was purified by flash chromatography on silica gel to give S4 (1.75 g, 42% yield) as yellow solid. $^1$H NMR (400 MHz, CDCl$_3$) δ 7.72 (s, $^1$H), 7.50 (s, $^1$H), 7.38 (s, $^1$H), 7.21 (t, J=7.7 Hz, $^1$H), 6.94 (d, J=7.5 Hz, $^1$H), 6.89 (s, $^1$H), 6.77 (d, J=7.9 Hz, $^1$H), 5.09 (s, $^1$H), 3.80 (s, $^3$H), 3.56 (d, J=9.7 Hz, $^1$H), 2.70–2.57 (m, $^2$H), 2.22–2.10 (m, $^2$H), 1.71–1.60 (m, $^3$H), 1.58–1.50 (m, $^1$H), 1.44–1.36 (m, $^1$H), 1.12 (d, J=9.9 Hz, 1H).

## Methyl 2-chloro-6-(3-((2,2-dimethyl-4-oxo-3,8,11-trioxa-5-azatridecan-13-yl)oxy)phenyl)-5,6,6a,7,8,9,10,10a-octahydro-7,10-methanophenanthridine-4-carboxylate (S6)

S4 (500 mg, 1.3 mmol), tert-butyl (2-(2-(2-hydroxyethoxy)ethoxy)ethyl)carbamate S5 (389 mg, 1.56 mmol) and triphenylphosphine (511 mg, 1.95 mmol) were dissolved in THF (20 mL) in a 100 mL flask. The reaction mixture was cooled to 0°C with an ice water bath. Diisopropyl azodicarboxylate (DIAD, 394 mg, 1.95 mmol) was added dropwise, after which the reaction mixture was allowed to warm to room temperature and stirred for 16 hr. The mixture was quenched with aqueous NH$_4$Cl solution (50 mL). The aqueous layer was extracted with ethyl acetate (100 mL × 2), and the combined organic layers were washed with brine (100 mL × 2), dried over anhydrous Na$_2$SO$_4$, filtered, and concentrated in vacuo. The crude residue was purified by flash chromatography on silica gel to give S6 (688 mg, 86% yield) as yellow solid. $^1$H NMR (400 MHz, CDCl$_3$) δ 7.72 (dd, J=2.5, 0.8 Hz, $^1$H), 7.50 (s, $^1$H), 7.38 (s, $^1$H), 7.26–7.21 (m, $^1$H), 7.01–6.95 (m, $^2$H), 6.86 (dd, J=8.2, 1.6 Hz, $^1$H), 5.03 (s, $^1$H), 4.18–4.14 (m, $^2$H), 3.89–3.85 (m, $^2$H), 3.80 (s, $^3$H), 3.75–3.70 (m, $^2$H), 3.67–3.63 (m, $^2$H), 3.58–3.53 (m, $^3$H), 3.35–3.29 (m, $^2$H), 2.67–2.60 (m, $^2$H), 2.20–2.12 (m, $^2$H), 1.65–1.60 (m, $^4$H), 1.44–1.41 (m, $^{10}$H), 1.13 (d, J=10.2 Hz, $^1$H).

## Methyl 2-chloro-6-(3-(2-(2-(2-(2-((S)-4-(4-chlorophenyl)–2,3,9-trimethyl-6H-thieno[3,2 f][1,2,4]triazolo[4,3 a][1,4]diazepin-6-yl)acetamido)ethoxy)ethoxy)ethoxy)phenyl)-5,6,6a,7,8,9,10,10a-octahydro-7,10-methanophenanthridine-4-carboxylate (7)

S6 (92 mg, 0.15 mmol) was dissolved in 4 M HCl in dioxane (5 mL), which was then stirred at room temperature for 2 hr. The mixture was then concentrated under reduced pressure. To the solution of the above residue and (+)-JQ1 carboxylic acid (40 mg, 0.1 mmol) in DMF (5 mL) was added diisopropylethylamine (77.4 mg, 0.6 mmol) and HATU (57 mg, 0.15 mmol). The mixture was stirred at

25°C for 12 hr, then diluted with 1 M aqueous HCl (20 mL) and extracted with ethyl acetate (20 mL × 2). The combined organic layers were washed with aqueous NaHCO$_3$ solution and brine (20 mL × 2), dried over sodium sulfate, filtered, and concentrated. The crude residue was purified by preparative HPLC to give 7 (19 mg, 21% yield) as yellow solid. $^1$H NMR (400 MHz, CDCl$_3$) δ 7.71 (d, J=2.3 Hz, $^1$H), 7.44–7.32 (m, 6 H), 7.22 (dd, J=7.9, 1.9 Hz, $^1$H), 6.97 (d, J=7.5 Hz, $^2$H), 6.86 (d, J=8.0 Hz, $^1$H), 4.78 (t, J=7.0 Hz, $^1$H), 4.17 (t, J=4.6 Hz, $^2$H), 3.90 (d, J=2.9 Hz, $^2$H), 3.79 (s, $^3$H), 3.78–3.74 (m, $^2$H), 3.73–3.69 (m, $^2$H), 3.68–3.60 (m, $^2$H), 3.59–3.49 (m, $^4$H), 3.48–3.42 (m, $^1$H), 2.82 (s, $^3$H), 2.65 (d, J=8.9 Hz, $^1$H), 2.61 (d, J=3.4 Hz, $^1$H), 2.44 (s, $^3$H), 2.19 (t, J=9.1 Hz, $^1$H), 2.15–2.10 (m, $^1$H), 1.69 (s, $^3$H), 1.67–1.58 (m, $^2$H), 1.57–1.47 (m, $^1$H), 1.45–1.36 (m, $^1$H), 1.26–1.16 (m, $^1$H), 1.12 (d, J=10.1 Hz, $^1$H). $^{13}$C NMR (101 MHz, CDCl$_3$) δ 171.03, 167.97, 159.05, 150.68, 149.16, 145.30, 138.25, 135.10, 133.49, 133.17, 131.85, 131.25, 131.11, 130.53, 130.32, 129.79, 129.77, 129.16, 128.04, 121.50, 120.57, 120.51, 114.00, 113.97, 112.76, 70.89, 70.53, 69.88, 69.53, 67.53, 59.73, 53.66, 52.75, 51.91, 43.96, 43.63, 40.09, 40.01, 37.79, 34.02, 29.77, 29.14, 14.45, 13.35, 11.30. HRMS calculated for C$_{47}$H$_{51}$Cl$_2$N$_6$O$_6$S (M$^+$+H) 897.2968, found 897.3001.

**Appendix 1—scheme 2.** Synethic scheme of the synthesis of Compound 8.

## Methyl 2-chloro-6-(3-(2-(piperidin-4-yl)ethoxy)phenyl)-5,6,6a,7,8,9,10,10a-octahydro-7,10-methanophenanthridine-4-carboxylate (S8)

S4 (766 mg, 2 mmol), 2-(piperidin-4-yl)ethan-1-ol S7 (307 mg, 2.4 mmol), and triphenylphosphine (786 mg, 3 mmol) were dissolved in THF (25 mL) in a 100 mL flask. The reaction mixture was cooled to 0°C with an ice water bath. DIAD (606 mg, 3 mmol) was added dropwise, after which the reaction mixture was allowed to warm to room temperature and stirred for 16 hr. The mixture was quenched with aqueous NH$_4$Cl solution (50 mL). The aqueous layer was extracted with ethyl acetate (40 mL × 2), and the combined organic layers were washed with brine (40 mL × 2), dried over anhydrous Na$_2$SO$_4$, filtered, and concentrated in vacuo. The crude residue was purified by flash chromatography on silica gel to give S8 (574 mg, 58% yield) as yellow solid. $^1$H NMR (400 MHz, CDCl$_3$) δ 7.75 (dd, J=2.5, 0.8 Hz, 1H), 7.41 (dd, J=2.4, 1.2 Hz, $^1$H), 7.28–7.25 (m, $^1$H), 7.00 (d, J=7.7 Hz, $^1$H), 6.99–6.96 (m, $^1$H), 6.84 (dd, J=8.2, 1.9 Hz, $^1$H), 4.04 (t, J=5.8 Hz, $^2$H), 3.83 (s, $^3$H), 3.59 (d, J=9.9 Hz, $^1$H), 3.45 (d, J=12.0 Hz, $^2$H), 3.00–2.85 (m, $^2$H), 2.70–2.62 (m, $^2$H), 2.22 (t, J=9.4 Hz, $^1$H), 2.17 (d, J=3.8 Hz, $^1$H), 2.00 (d, J=13.9 Hz, $^2$H), 1.95–1.87 (m, $^1$H), 1.86–1.79 (m, $^2$H), 1.73–1.53 (m, $^5$H), 1.49–1.40 (m, $^1$H), 1.30–1.20 (m, $^1$H), 1.16 (d, J=10.2 Hz, $^1$H).

## Methyl 6-(3-(2-(1-((1-(*tert*-butoxycarbonyl)piperidin-4-yl)methyl)piperidin-4-yl)ethoxy)phenyl)-2-chloro-5,6,6a,7,8,9,10,10a-octahydro-7,10-methanophenanthridine-4-carboxylate (S10)

To a solution of S8 (99 mg, 0.2 mmol) in methanol (3 mL) was added sodium acetate (29 mg, 0.4 mmol), acetic acid (10 μL, 0.2 mmol), and sodium cyanoborohydride (33 mg, 0.5 mmol) at 0°C. Then, tert-butyl 4-formylpiperidine-1-carboxylate S9 (56 mg, 0.3 mmol) was added, and the mixture was stirred at room temperature for 16 hr. The reaction was concentrated under reduced pressure. The residue was purified by flash chromatography on silica gel to give S10 (101 mg, 71% yield) as yellow solid. $^1$H NMR (400 MHz, CDCl$_3$) δ 7.72 (d, J=1.7 Hz, $^1$H), 7.48 (s, $^1$H), 7.39 (s, $^1$H), 7.26–7.22 (m, $^1$H), 6.97 (d, J=7.7 Hz, $^1$H), 6.95–6.92 (m, $^1$H), 6.82 (dd, J=8.2, 1.9 Hz, $^1$H), 4.20–4.06 (m, $^2$H), 4.01

(t, $J$=5.8 Hz, [2]H), 3.80 (s, [3]H), 3.56 (d, $J$=9.8 Hz, [1]H), 3.27–3.08 (m, [2]H), 2.72–2.61 (m, [4]H), 2.19 (t, $J$=9.4 Hz, [1]H), 2.14 (d, $J$=3.8 Hz, [1]H), 1.90–1.84 (m, [2]H), 1.80–1.70 (m, [5]H), 1.70–1.50 (m, [10]H), 1.45 (s, [9]H), 1.41–1.37 (m, [1]H), 1.24–1.19 (m, [1]H), 1.18–1.15 (m, [1]H), 1.15–1.10 (m, [2]H).

## Methyl 2-chloro-6-(3-(2-(1-((1-(2-(($S$)–4-(4-chlorophenyl)-2,3,9-trimethyl-6$H$-thieno[3,2 $f$][1,2,4]triazolo[4,3 $a$][1,4]diazepin-6-yl)acetyl)piperidin-4-yl)methyl)piperidin-4-yl)ethoxy)phenyl)-5,6,6a,7,8,9,10,10a-octahydro-7,10-methan-ophenanthridine-4-carboxylate (8)

S10 (72 mg, 0.1 mmol) was dissolved in 4 M HCl in dioxane (3 mL), which was then stirred at room temperature for 2 hr. The mixture was then concentrated under reduced pressure. To the solution of the above residue and (+)-JQ1 carboxylic acid (49 mg, 0.12 mmol) in DMF (5 mL) was added diisopropylethylamine (65 mg, 0.5 mmol) and HATU (57 mg, 0.15 mmol). The mixture was stirred at room temperature for 12 hr, then diluted with 1 M aqueous HCl (20 mL) and extracted with ethyl acetate (20 mL × 2). The combined organic layers were washed with aqueous NaHCO$_3$ solution and brine (20 mL × 2), dried over sodium sulfate, filtered, and concentrated. The crude residue was purified by preparative HPLC to give 8 (15 mg, 15% yield) as yellow solid. [1]H NMR (400 MHz, CDCl$_3$) δ 7.72 (s, [1]H), 7.43–7.35 (m, [5]H), 7.26–7.22 (m, [1]H), 6.98 (d, $J$=7.7 Hz, [1]H), 6.94 (s, [1]H), 6.83–6.80 (m, [1]H), 4.91–4.83 (m, [1]H), 4.64–4.53 (m, [1]H), 4.29–4.13 (m, [1]H), 4.06–3.98 (m, [2]H), 3.97–3.82 (m, [1]H), 3.80 (s, [3]H), 3.78–3.68 (m, [2]H), 3.57 (d, $J$=9.9 Hz, [1]H), 3.35–3.14 (m, [2]H), 3.06–2.90 (m, [2]H), 2.82 (s, [3]H), 2.78–2.57 (m, [5]H), 2.44 (s, [3]H), 2.30–2.16 (m, [2]H), 2.15–2.12 (m, [1]H), 2.06–1.93 (m, [3]H), 1.90–1.74 (m, [6]H), 1.70 (s, [3]H), 1.69–1.59 (m, [2]H), 1.59–1.50 (m, [1]H), 1.43–1.38 (m, [1]H), 1.31–1.18 (m, [2]H), 1.14 (d, $J$=10.1 Hz, [1]H). [13]C NMR (101 MHz, CDCl$_3$) δ 168.63, 168.07, 168.04, 165.37, 164.99, 160.95, 160.56, 159.09, 155.54, 149.33, 145.47, 137.76, 135.73, 133.50, 133.09, 131.78, 131.67, 130.43, 130.34, 130.07, 129.77, 129.14, 129.09, 128.05, 121.40, 120.53, 114.28, 113.72, 113.68, 112.66, 64.60, 63.90, 59.74, 55.09, 54.85, 54.54, 52.86, 51.89, 46.25, 43.97, 43.59, 42.26, 39.99, 34.92, 34.57, 34.03, 32.37, 30.74, 30.70, 30.35, 29.79, 29.15, 29.02, 14.47, 13.35, 11.16. HRMS calcd for C$_{54}$H$_{62}$Cl$_2$N$_7$O$_4$S (M$^+$+H) 974.3961, found 974.3993.

**Appendix 1—scheme 3.** Synethic scheme of the synthesis of Compound 9.

## tert-Butyl (4'-((3-formylphenoxy)methyl)-[1,1'-biphenyl]-4-yl)carbamate (S12)

S3 (244 mg, 2 mmol), tert-butyl (4'-(hydroxymethyl)-[1,1'-biphenyl]-4-yl)carbamate S11 (1.3 g, 2.4 mmol), and triphenylphosphine (629 mg, 2.4 mmol) were dissolved in THF (10 mL) in a 50 mL flask. The reaction mixture was cooled to 0°C with an ice water bath. DIAD (485 mg, 2.4 mmol) was added dropwise, after which the reaction mixture was allowed to warm to room temperature and stirred for 16 hr. The mixture was quenched with aqueous NH$_4$Cl solution (50 mL). The aqueous layer was extracted with ethyl acetate (40 mL × 2), and the combined organic layers were washed with brine (40 mL × 2), dried over anhydrous Na$_2$SO$_4$, filtered, and concentrated in vacuo. The crude residue was purified by flash chromatography on silica gel to give S12 (460 mg, 57% yield) as white solid. [1]H NMR (500 MHz, CDCl$_3$) δ 9.98 (s, [1]H), 7.59 (d, $J$=8.1 Hz, [2]H), 7.53 (d, $J$=8.5 Hz, [2]H), 7.51–7.48 (m, [4]H), 7.46 (t, $J$=6.8 Hz, [2]H), 7.29–7.26 (m, [1]H), 6.55 (s, [1]H), 5.16 (s, [2]H), 1.54 (s, [9]H).

## Methyl 6-(3-((4'-((*tert*-butoxycarbonyl)amino)-[1,1'-biphenyl]-4-yl) methoxy)phenyl)-2-chloro-5,6,6a,7,8,9,10,10a-octahydro-7,10-methanophenanthridine-4-carboxylate (S13)

Methyl 2-amino-5-chlorobenzoate S1 (185 mg, 1 mmol), norbornene S2 (141 mg, 1.5 mmol), and S12 (443 mg, 1.1 mmol) were dissolved in DCM (5 mL) in a 50 mL Ace pressure tube. After purging with argon, boron trifluoride etherate (28 mg, 0.2 mmol) was added. The tube was sealed instantly, and the reaction mixture was heated at 40°C for 16 hr. 1 mL of methanol was added to the mixture to quench the reaction. After concentrating in vacuo, the crude residue was purified by flash chromatography on silica gel to give S13 (193 mg, 29% yield) as yellow solid. $^1$H NMR (400 MHz, CDCl$_3$) δ 7.72 (d, $J$=1.9 Hz, $^1$H), 7.58 (d, $J$=8.2 Hz, $^2$H), 7.53 (d, $J$=8.5 Hz, $^3$H), 7.49 (d, $J$=8.2 Hz, $^2$H), 7.43 (d, $J$=8.5 Hz, $^2$H), 7.38 (s, $^1$H), 7.30–7.23 (m, 2 H), 7.06 (s, 1 H), 7.00 (d, $J$=7.7 Hz, 1 H), 6.93 (dd, $J$=8.2, 2.0 Hz, $^1$H), 6.55 (s, $^1$H), 5.10 (s, $^2$H), 3.80 (s, $^3$H), 3.59 (d, $J$=9.8 Hz, $^1$H), 2.69–2.59 (m, $^2$H), 2.22–2.12 (m, $^2$H), 1.72–1.56 (m, $^4$H), 1.54 (s, $^9$H), 1.23–1.16 (m, $^1$H), 1.12 (d, $J$=10.2 Hz, $^1$H).

## Methyl 2-chloro-6-(3-((4'-(2-((*S*)-4-(4-chlorophenyl)-2,3,9-trimethyl-6*H*-thieno[3,2 *f*][1,2,4]triazolo[4,3 *a*][1,4]diazepin-6-yl) acetamido)-[1,1'-biphenyl]-4-yl)methoxy)phenyl)--5,6,6a,7,8,9,10,10a-octahydro-7,10-methanophenanthridine-4-carboxylate (9)

S13 (66 mg, 0.1 mmol) was dissolved in 4 M HCl in dioxane (3 mL), which was then stirred at room temperature for 2 hr. The mixture was then concentrated under reduced pressure. To the solution of the above residue and (+)-JQ1 carboxylic acid (49 mg, 0.12 mmol) in DMF (5 mL) was added diisopropylethylamine (65 mg, 0.5 mmol) and HATU (57 mg, 0.15 mmol). The mixture was stirred at room temperature for 12 hr, then diluted with 1 M aqueous HCl (20 mL) and extracted with ethyl acetate (20 mL × 2). The combined organic layers were washed with aqueous NaHCO$_3$ solution and brine (20 mL × 2), dried over sodium sulfate, filtered, and concentrated. The crude residue was purified by preparative HPLC to give 9 (36 mg, 38% yield) as yellow solid. $^1$H NMR (400 MHz, CDCl$_3$) δ 8.81 (s, $^1$H), 7.72 (d, $J$=1.7 Hz, $^1$H), 7.64 (d, $J$=8.6 Hz, $^2$H), 7.60–7.55 (m, $^4$H), 7.50 (d, J=8.2 Hz, $^2$H), 7.42 (d, $J$=8.5 Hz, $^2$H), 7.39–7.36 (m, $^3$H), 7.29–7.26 (m, $^2$H), 7.06 (s, $^1$H), 7.00 (d, $J$=7.6 Hz, $^1$H), 6.93 (dd, $J$=8.2, 1.9 Hz, $^1$H), 5.10 (s, $^2$H), 4.81 (t, $J$=6.8 Hz, $^1$H), 3.80 (s, $^3$H), 3.61–3.58 (m, $^3$H), 2.80 (s, $^3$H), 2.68–2.60 (m, $^2$H), 2.45 (s, $^3$H), 2.20 (t, $J$=9.3 Hz, $^1$H), 2.14–2.11 (m, $^1$H), 1.73–1.70 (m, $^3$H), 1.68–1.60 (m, $^2$H), 1.55–1.50 (m, $^1$H), 1.43–1.37 (m, $^1$H), 1.22–1.17 (m, $^1$H), 1.12 (d, $J$=10.3 Hz, $^1$H). $^{13}$C NMR (101 MHz, CDCl$_3$) δ 168.66, 167.99, 159.14, 150.57, 149.35, 145.46, 140.51, 140.36, 137.91, 137.21, 137.18, 135.99, 135.64, 133.49, 132.64, 131.63, 131.27, 130.43, 130.24, 130.22, 130.18, 129.79, 129.14, 128.26, 128.09, 128.06, 127.76, 127.20, 121.34, 120.50, 114.26, 114.23, 112.67, 69.92, 59.71, 54.30, 52.85, 51.90, 43.98, 43.69, 40.06, 39.83, 34.04, 29.80, 29.20, 14.57, 13.36, 11.55. HRMS calculated for C$_{54}$H$_{49}$Cl$_2$N$_6$O$_4$S (M$^+$+H) 947.2908, found 947.2921.

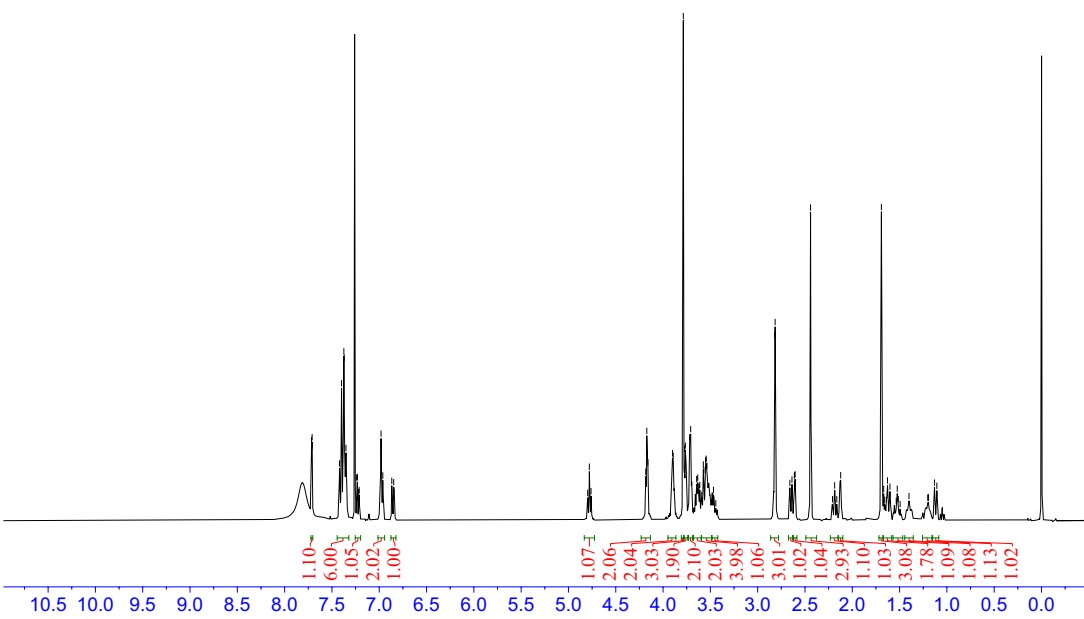

**Appendix 1—figure 1.** $^1$H NMR spectra of **7** in CDCl$_3$.

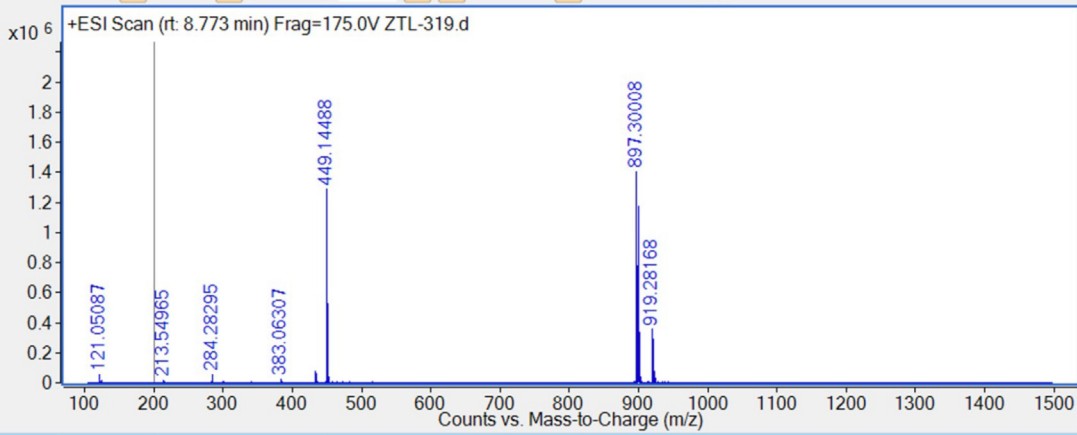

**Appendix 1—figure 2.** HRMS spectra of **7**.

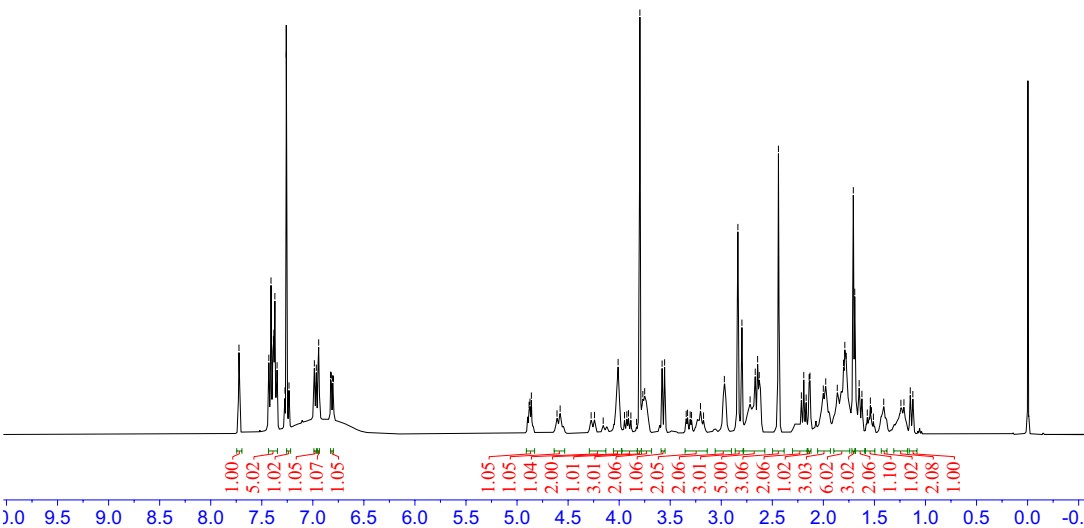

**Appendix 1—figure 3.** ${}^{1}$H NMR spectra of **8** in CDCl$_3$.

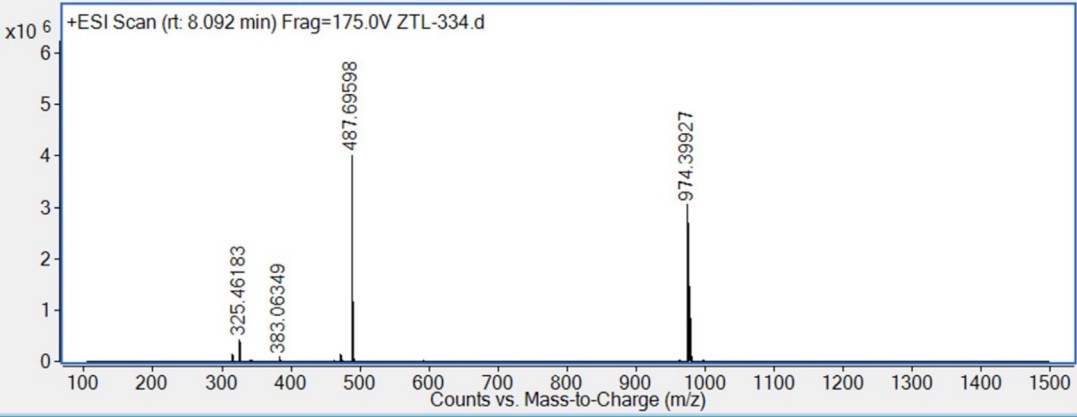

**Appendix 1—figure 4.** HRMS spectra of **8**.

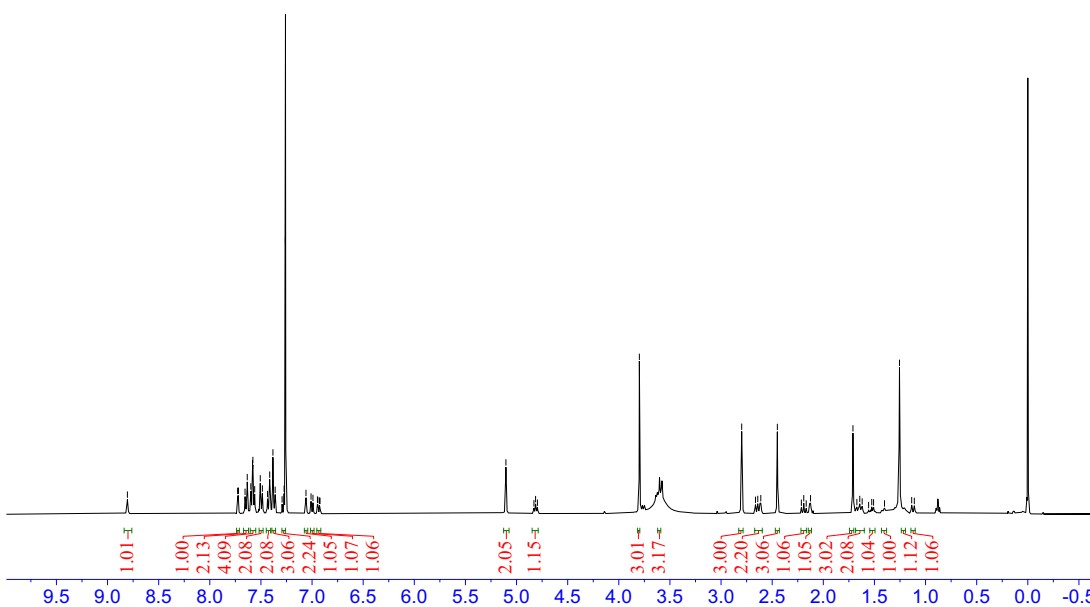

**Appendix 1—figure 5.** $^1$H NMR spectra of **9** in CDCl$_3$.

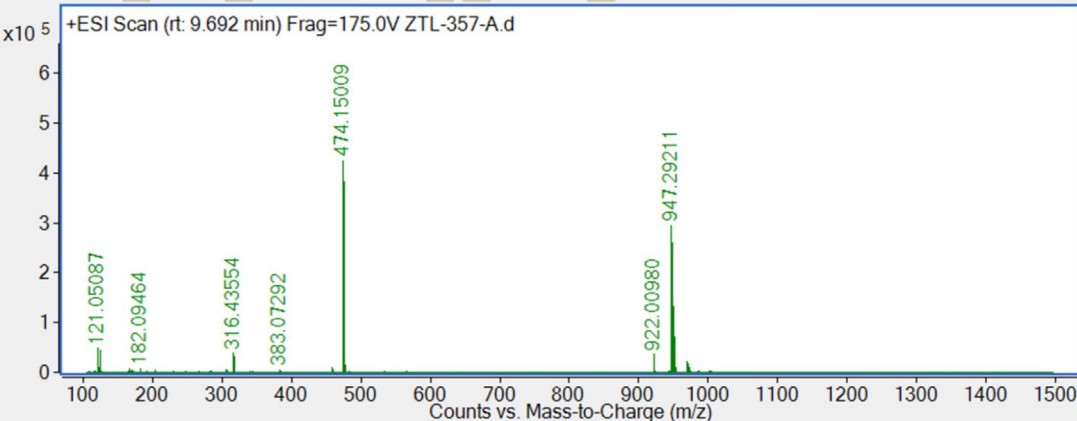

**Appendix 1—figure 6.** HRMS spectra of **9**.

