## [Editor Report · eLife Assessment]

This **valuable** study aims to advance the toolkit of small molecules used for approaches to targeted protein degradation for research and therapeutic applications. The authors provide **solid** data demonstrating the use of a high-throughput screen of small molecules to target a specific E3 ligase, KLHDC2 (Kelch-like homology domain containing protein 2); the resulting compounds then form the basis for new PROTAC (proteolysis targeting chimera) reagents. The strength of the work lies in expanding the PROTAC reagent inventory. The current work would be strengthened further by confirming that the PROTAC's activity is dependent on KLHDC2 and by a more thorough examination of off-target effects in cellular applications.

---

## [Referee Report · Reviewer #1 (Public review)]

Summary:

The manuscript "Targeted Protein Degradation by KLHDC2 Ligands Identified by High Throughput Screening" by Zhou, H. et al. describes the development of a high-throughput FP-based screen and the identification of a KLHDC2 ligand from a small molecule library. A counter screen and other filtering criteria led to the identification of lead compounds that contained a tetrahydroquinoline scaffold. Commercially available analogs (52 compounds) that shared this scaffold were characterized by a KLHDC2 competitive binding assay. Optimized compounds were obtained that demonstrated improved potency and increased binding affinity by SPR. Docking of a lead candidate (compound 6) suggested it bound at a distal lipophilic site within the SelK binding pocket of KLHDC2. Based on this model, the authors then synthesized PROTACs that linked the KLHDC2 binder to a BRD4-binding molecule, JQ1. These PROTAC candidates possessed different linker configurations, and PROTAC 8 was able to cause BRD4 degradation in cells, with a half-maximal degradation concentration (DC50) of 80 nM. The authors demonstrate the identification and characterization of small-molecule KLHDC2 ligands that can be used to generate PROTACs that result in BRD4 degradation in cells.

Strengths:

The study by Zhou, H. et al. expands the E3 ligase toolkit by targeting KLHDC2 to identify ligands for PROTAC development, which has predominantly relied on VHL and CRBN. This was accomplished using a described FP-based high-throughput screening strategy (high Z' values in 1536 well format). Both target-specific and counter-specific assays were performed, along with subsequent stringent follow-up assays designed to address non-specific binding/specificity concerns. Label-free direct binding validations by SPR were used to determine binding affinity/kinetics. A strength of the study is the characterization of the interaction between candidate compounds and KLHDC2 versus related KEAP1.

Structural insight into the potential mode of binding was inferred by computational docking studies of the newly discovered KLHDC2 ligands. This was performed to identify where the identified scaffolds could be modified by linker incorporation for the design of PROTACs. The computational predictions were evaluated by linking a solvent-exposed site on the KLHDC2 ligand to JQ1. Three linkers were tested, and two compounds were found to result in BRD4 degradation in cells by HiBiT degradation assay and western blot. These findings demonstrate the feasibility of these compounds for the design of PROTAC-based degraders.

The authors present compelling KLHDC2 binding data for their lead compounds and demonstrate degradation of a target using a PROTAC strategy. Accordingly, the screening approach and compounds identified are likely to be of interest to the field and are likely to be generalizable to other PROTAC targets of interest.

Weaknesses:

The specificity of compounds for KLHDC2 was assessed by using a counter screen against KEAP1 and in vitro binding assays. However, off-target effects might occur in a cellular context, which weren't fully explored in the study. Notably, the authors do not demonstrate that the degradation induced by their PROTACs in cells is KLHDC2-dependent. A requirement for KLHDC2-mediated degradation could be evaluated, for example, by using knockout/knockdown of KLHDC2, or other means, to demonstrate specificity. Addressing specificity is deemed important to evaluate the proposed PROTAC mechanism of action in a cellular context that results in the degradation of BRD4. Specificity is important when considering the utility of these new compounds for PROTAC design.

Additional rationale behind the selection of linkers used to generate candidate PROTACs would be informative and would benefit from additional discussion and/or citation. The reasons for the lack of activity, such as for compound 9, were not fully explored or discussed, such as whether complex assembly is potentially affected by linker choice. Perhaps related to this point, the authors note that a trifluoromethoxy group increased the binding affinity of compound 6. However, the subsequent docking analysis revealed this moiety to be solvent-exposed. The relationship between this site of functionalization, linker selection, and the resulting binding affinity or effect on DC50 was not clear and/or could be developed further.

Minor issues related to the presentation of the manuscript include sections that would benefit from either additional citation and/or description, such as the KI-696 inhibitor used and the BRD4 HiBiT degradation assay that was used to assess PROTAC potency. Figure captions should be reviewed to ensure that the number of independent experiments is indicated, and what data points and error bars represent, as these are not indicated in several figures. BRD4 levels were quantified in 4E; however, error/reproducibility (n) is not indicated.

---

## [Referee Report · Reviewer #2 (Public review)]

PROTACs are a class of small molecules that induce an interaction between a target protein and a ubiquitin ligase, thereby leading to the target protein's ubiquitination and subsequent proteasomal degradation. Given that the vast majority of PROTACs rely on the cereblon and VHL ubiquitin ligases, a major goal within this field has been to identify and develop ligands for additional ubiquitin ligases, in particular those whose expression affords tissue or subcellular specificity or those whose structure allows them to degrade targets that are otherwise incompatible with cereblon or VHL.

In this work, Zhou and colleagues from the Bollong group at Scripps utilize a high-throughput fluorescence polarization screen of >350,000 compounds to identify and optimize a novel ligand for KLHDC2, a ubiquitin ligase which had previously been discovered to be capable of proximity-induced degradation of target proteins. Zhou et al go on to show that this ligand can be used as the basis for PROTACs capable of degrading BRD4 in a cell line. Of note, prior to this paper, three other groups had also developed ligands to KLHDC2 and used them to generate active PROTACs. Interestingly, docking studies by Zhou suggest that their compound may bind to a different region of the KLHDC2's kelch domain.

The major strengths of this work are its brevity and the clarity of the writing and figures. Their claim that they have discovered a ligand for KLHDC2, which can be used to develop BRD4-degrading PROTACs, is well-supported by their findings from the screen, SPR, and cellular assays. The weakness of the work then, is not so much relevant to the paper at hand but rather stems from the fact that their story leaves me wanting to know more. Indeed, there are a number of interesting experiments that we need as a field in order to assess (1) how generalizable their findings are across cell lines and targets, and (2) how this new KLHDC2 ligand stacks up against the other recently discovered ligands for KLDHC2 as well as the existing standards, cereblon and VHL.

Nonetheless, Zhou and colleagues provide a valuable addition to the emerging repertoire of KLHDC2 ligands, and I'm certain that with time, we will come to understand what ligands work best for KLHDC2-based PROTACs and how they compare to the growing set of ubiquitin ligases in our armamentarium.